# A Bayesian Approach for Designing Experiments Based on Information Criteria to Reduce Epistemic Uncertainty of Fuel Fracture During Loss-of-Coolant Accidents †

Shusuke Hamaguchi, Takafumi Narukawa *  and Takashi Takata *

Department of Nuclear Engineering and Management, Graduate School of Engineering, The University of Tokyo, Tokyo 113-8656, Japan; hamaguchi@nse.t.u-tokyo.ac.jp
* Correspondence: narukawa@n.t.u-tokyo.ac.jp (T.N.); takata_t@n.t.u-tokyo.ac.jp (T.T.)
† This article is a revised and expanded version of a paper entitled Hamaguchi, S.; Narukawa, T.; Takata T. A Bayesian approach for designing experiments based on information criteria to reduce prediction uncertainty of fuel fracture during LOCA. In Proceedings of the Probabilistic Safety Assessment and Management & Asian Symposium on Risk Assessment and Management (PSAM 17 & ASRAM 2024), Sendai International Center, Sendai, Miyagi, Japan, 7–11 October 2024.

## Abstract

In probabilistic risk assessment (PRA), the fracture limit of fuel cladding tubes under loss-of-coolant accident conditions plays a critical role in determining the core damage, highlighting the need for accurate modeling of cladding tube fracture behavior. However, for high-burnup cladding tubes, it is often infeasible to conduct extensive experiments due to limited material availability, high costs, and technical constraints. These limitations make it difficult to acquire sufficient data, leading to substantial epistemic uncertainty in fracture modeling. To enhance the realism of PRA results under such constraints, it is essential to develop methods that can effectively reduce epistemic uncertainty using limited experimental data. In this study, we propose a Bayesian approach for designing experimental conditions based on a widely applicable information criterion (WAIC) in order to effectively reduce the uncertainty in the prediction of fuel cladding tube fracture with limited data. We conduct numerical experiments to evaluate the effectiveness of the proposed method in comparison with conventional approaches based on empirical loss and functional variance. Two cases are considered: one where the true and predictive models share the same mathematical structure (Case 1) and one where they differ (Case 2). In Case 1, the empirical loss-based design performs best when the number of added data points is fewer than approximately 10. In Case 2, the WAIC-based design consistently achieves the lowest Bayes generalization loss, demonstrating superior robustness in situations where the true model is unknown. These results indicate that the proposed method enables more informative experimental designs on average and contributes to the effective reduction in epistemic uncertainty in practical applications.

**Keywords:** loss-of-coolant accident; fuel cladding tube; fracture limit; information criteria; Bayesian update

## 1. Introduction

In the field of probabilistic risk assessment (PRA), reducing epistemic uncertainty has been a central challenge, particularly in the evaluation of rare or high-consequence safety-related events [1–3]. Epistemic uncertainty arises from incomplete knowledge about the system or environment under analysis and is typically classified into three categories:

- Parameter uncertainty, which originates from insufficient or imprecise knowledge about the true values of model parameters. This may result from limited, noisy, or biased data, as well as from an incomplete understanding of the physical processes that the parameters represent.
- Model uncertainty, which reflects the possibility that the mathematical structure of the model does not fully capture the true behavior of the physical system, due to simplifying assumptions or incomplete theoretical understanding.
- Incompleteness uncertainty, which refers to the absence of relevant variables, mechanisms, or interactions in the model—either because they are known but excluded (known unknowns) or because they have not yet been recognized (unknown unknowns).

This is distinct from aleatory uncertainty, which stems from inherent variability in natural processes—such as the randomness of material failure or the stochastic timing of events—and is considered irreducible by further information. While aleatory uncertainty is irreducible, epistemic uncertainty can, in principle, be reduced through additional data collection, improved modeling, and expanded knowledge and insight. In nuclear safety research, epistemic uncertainty plays a particularly important role in assessing fuel cladding behavior during a loss-of-coolant accident (LOCA), where the fracture limit is a critical threshold influencing accident progression and overall risk.

In particular, the fracture limit of fuel cladding tubes under LOCA conditions serves as a critical threshold for judging the core damage in many PRA models [4–6]. This limit is determined by whether the stresses generated during LOCA—such as thermal stress from rapid cooling during reflood and tensile stress from structural constraint—exceed the mechanical strength of the cladding. However, both the stress evaluation and strength estimation involve considerable uncertainty. The stresses are difficult to predict precisely due to the complex thermal–mechanical environment, and the strength is affected by multiple phenomena, including oxidation, hydrogen embrittlement, and wall thinning due to ballooning. These factors lead to epistemic uncertainty in the fracture limit. Therefore, reducing this uncertainty is essential for enhancing the realism and credibility of PRA results.

However, conducting extensive LOCA-simulated experiments, especially for high-burnup cladding tubes, is often infeasible due to limited material availability and high associated costs [7,8]. These constraints make it difficult to build statistically rich datasets for precise fracture modeling and fracture limit estimation. Therefore, it is essential to employ experimental design methodologies [9–14] that can maximize the information gained from each individual experiment, thereby reducing epistemic uncertainty in fracture modeling under data-scarce conditions.

One promising approach is the use of Bayesian optimal experimental design [9,10], which incorporates uncertainty in both model parameters and predictions and is especially well-suited for data-constrained safety assessment problems. Among its many applications across disciplines, a notable early example in the nuclear domain is a study by Yamaguchi et al. [15], which applied entropy-based Bayesian experimental design to seismic PRA by optimizing test conditions for component fragility modeling. This pioneering study demonstrated the practical value of Bayesian design in reducing uncertainty for rare, safety-critical events in nuclear engineering. Their method aimed to minimize information entropy (empirical loss) in Bayesian models, thereby reducing the Kullback–Leibler (KL) divergence between the true and predictive distributions [16].

Building upon this foundation, our previous work extended the concept by proposing a design criterion based on functional variance, which quantifies the variability in model output due to parameter uncertainty [17]. While both entropy-based and functional variance-based approaches offer useful heuristics for guiding experimental design, they

do not provide a principled mechanism for minimizing the KL divergence between the true and predictive distributions. Since KL divergence directly quantifies the discrepancy between what the model predicts and what is true, minimizing it is fundamental to reducing epistemic uncertainty. Without explicitly targeting KL divergence, these heuristic approaches may lead to suboptimal experimental conditions—particularly under limited data or model misspecification—where alignment between the predictive model and the underlying system becomes critical for reliable inference.

To address this limitation, we propose a Bayesian experimental design method based on the widely applicable information criterion (WAIC) [18], which asymptotically approximates the KL divergence between the true model and the posterior predictive distribution, even in complex or singular models. Unlike entropy-based or functional variance-based criteria, WAIC provides an asymptotically unbiased estimator of the predictive accuracy, even in cases of model misspecification or limited sample size. To the best of our knowledge, this is the first study to apply WAIC as a basis for experimental design in the context of probabilistic fracture modeling for nuclear fuel cladding under LOCA conditions. This study applies the WAIC-based design method to a fracture probability estimation model for fuel cladding tubes under LOCA conditions [19]. Numerical experiments are conducted to demonstrate the effectiveness of the proposed method in reducing epistemic uncertainty with limited data, thereby contributing to more realistic PRA outcomes and improved safety decision-making.

The novelty of this study lies in three aspects: (1) the application of WAIC, a modern Bayesian information criterion, to experimental design problems in nuclear safety; (2) the demonstration of its effectiveness in reducing epistemic uncertainty for fracture prediction under LOCA conditions; and (3) the comparison with conventional approaches (e.g., entropy-based and functional variance-based designs), highlighting its superior robustness under data-scarce scenarios. These contributions aim to improve the reliability of PRA by enhancing the realism of input parameter estimation.

Although the present study focuses on experiment design under data-scarce conditions, it is complementary to efforts aimed at improving the fidelity of LOCA simulations and experiments through model enhancements and data assimilation frameworks (e.g., refs. [20–22]).

This article is a revised and expanded version of a paper entitled Hamaguchi et al. [23].

## 2. Reduction in Epistemic Uncertainty

WAIC has been developed as a measure of the prediction accuracy of Bayesian models [18]. The WAIC calculated for each experimental condition represents the prediction accuracy of the model for each condition, and this information can be used to determine the value of a new experiment. In this study, we propose a method to reduce the parameter uncertainty of the model, which is one of the epistemic uncertainties, by preferentially conducting experiments under experimental conditions where WAIC is large (where the model's prediction accuracy is low) and using the obtained data for Bayesian update of the model.

WAIC is defined as follows:

$$WAIC = T + \frac{V}{n} \tag{1}$$

$$V = \sum_{i=1}^{n} E_\omega[(\log p(y_i \mid \omega))^2] - E_\omega[(\log p(y_i \mid \omega))]^2 \tag{2}$$

$$T = -\frac{1}{n}\sum_{i=1}^{n} \log p^*(y_i) \tag{3}$$

$$p * (y_i) = \mathrm{E}_\omega[p(y_i|\boldsymbol{\omega})] \tag{4}$$

where $T$ is the empirical loss, $V$ is the functional variance, $y_i$ is a random variable, $\boldsymbol{\omega}$ is the vector of parameters, $p(y_i|\boldsymbol{\omega})$ is a Bayesian model, $\mathrm{E}_\omega[]$ is the expectation value over the posterior distribution of $\boldsymbol{\omega}$, $p * (y_\mathbf{i})$ is the posterior predictive distribution, and $n$ is the number of samples.

As shown in Equation (5), the average of WAIC asymptotically approaches the average of the Bayes generalization loss G [18], which represents the prediction accuracy of the model. Therefore, WAIC can be used as a measure of the prediction accuracy of Bayesian models.

$$E[G] = E[WAIC] + O(\frac{1}{n^2}) \tag{5}$$

$$S = -\int q(y) \log q(y) \tag{6}$$

$$KL = \int q(y) \log \frac{q(y)}{p^*(y)} dy \tag{7}$$

$$G = S + KL = -\int q(y) \log p * (y) dy \tag{8}$$

where $q(y)$ is the true model, E[] is the expected value, $S$ is the entropy, and $KL$ is the KL divergence between $q(y)$ and $p * (y)$.

In this study, we propose a method to reduce parameter uncertainty of the model by preferentially conducting experiments under experimental conditions where the WAIC is large (where the prediction accuracy of the model is low). The proposed method consists of the following three steps, as shown in Figure 1.

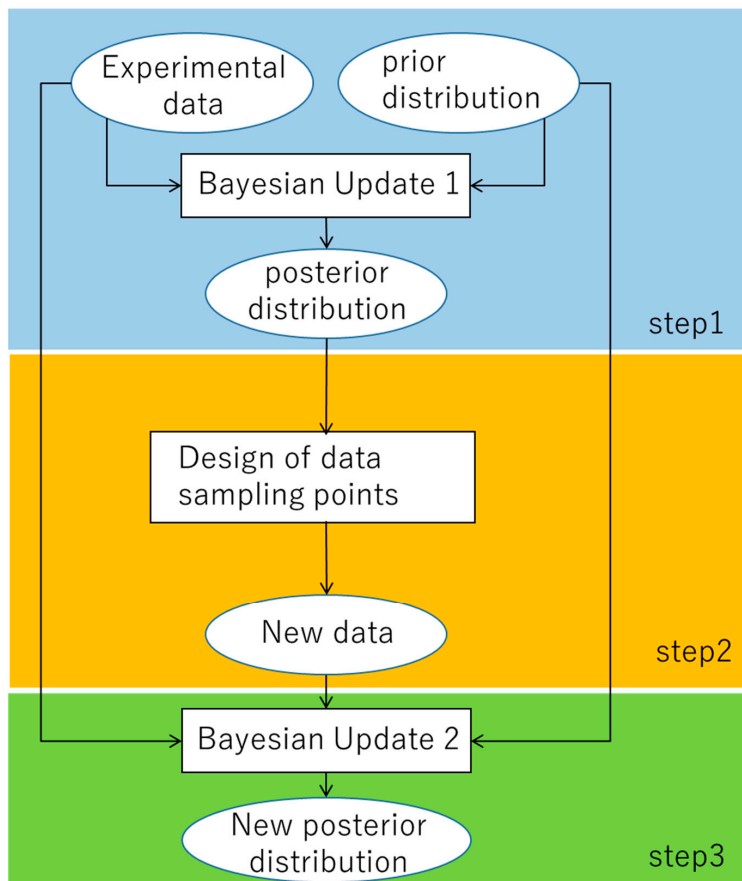

**Figure 1.** Flow of reduction in epistemic uncertainty.

[Step1] Bayesian update 1

The first Bayesian update is performed using the experimental data and prior distribution to obtain the posterior distribution.

[Step2] Design of data sampling points

WAIC at each point on the design space of the data sampling points is calculated from the posterior distribution of parameters, and the experiment is conducted at the data sampling point with a large WAIC value to obtain new data.

[Step3] Bayesian update 2

Then, the second Bayesian update is performed, including the newly added experimental data, and the posterior distribution of parameters is updated. In this way, a new posterior distribution with reduced parameter uncertainty is obtained.

## 3. Numerical Experiments

To evaluate the effectiveness of the proposed epistemic uncertainty reduction method, numerical experiments were conducted by applying the proposed method to a fracture probability estimation model [19]. The fracture probability estimation model provides an estimate of the fracture probability of a non-irradiated Zircaloy-4 cladding tube under LOCA conditions. This model uses equivalent cladding reacted (ECR), which quantifies the extent of oxidation and initial hydrogen concentration, representing the hydrogen content prior to the LOCA transient, as explanatory variables, since these are the dominant factors influencing cladding embrittlement and fracture under LOCA conditions.

This understanding is based on decades of experimental research identifying oxidation and hydrogenation as the primary mechanisms of cladding fracture [24–31]. Although the current numerical experiments focus on non-irradiated cladding, our previous study [31] has reported that the fracture limit is not significantly reduced even at burnups up to approximately 85 GWd/t. This suggests that the proposed method is also applicable to high-burnup cladding tubes and remains relevant for realistic safety evaluations.

In these experiments, a "true" model was assumed to generate binary fracture/non-fracture data. While such a true model is unobservable in actual applications, constructing a predictive model that approximates this unknown mechanism is a central challenge in safety modeling. To examine the performance of the proposed method under different modeling conditions, we considered two representative cases: Case 1, where the true and predictive models share the same mathematical structure, and Case 2, where they differ. These cases were selected to capture both ideal and more realistic situations, including model misspecification, which is frequently encountered in practice.

Although the experiments focus on these two cases, the proposed Bayesian design methodology is not limited to them. Rather, it is broadly applicable to other safety-critical systems where experimental data are limited and epistemic uncertainty is significant. The present evaluation aims to demonstrate the method's utility in typical scenarios, thereby supporting its generalizability.

*3.1. Case 1: True and Predictive Models Have the Same Mathematical Structure*

3.1.1. Model Definition

The true model is defined as follows, as in the previous study [11]:

$$Y \sim Bernoulli(P_{true}(Y = 1|\boldsymbol{X})) \tag{9}$$

$$P_{true} = \Phi(10 + 7\log(\frac{X_1}{100}) + 20\log(1 + \frac{X_2}{10,000})) \tag{10}$$

where $Y$ is LOCA-simulated test data binarized to 1 for fracture and 0 for non-fracture, *Bernoulli*() is Bernoulli distribution, $P_{true}$ is the fracture probability estimated by the true model, $X_1$ is an explanatory variable for ECR (%), $X_2$ is an explanatory variable for the initial hydrogen concentration (wtppm), and $\Phi$ is the cumulative distribution function of the standard normal distribution.

The predictive model is defined as follows:

$$Y \sim Bernoulli(P(Y = 1|\mathrm{X})) \tag{11}$$

$$P = \Phi(\omega_0 + \omega_1 \log(\frac{X_1}{100}) + \omega_2 \log(1 + \frac{X_2}{10,000})) \tag{12}$$

where $P$ is the fracture probability, estimated by the predictive model, and $(\omega_0, \omega_1, \omega_2)$ are parameters to be estimated.

Marginal prior distributions of these parameters are assumed to follow the following noninformative prior distribution:

$$\omega_k \sim Normal(0, 100) \ (k = 0, 1, 2) \tag{13}$$

3.1.2. Calculation Flow

Numerical experiments were conducted in the following steps (a) to (f).

(a) The design space of data sampling points was defined as a two-dimensional space consisting of ECR and initial hydrogen concentration. The design space consists of 403 sampling points in which ECR ranges from 10% to 40% in 1% increments, and the initial hydrogen concentration ranges from 0 wtppm to 1200 wtppm in 100 wtppm increments.

(b) Initial data were set as in the previous study [17], as shown in Figure 2.

(c) Bayesian inference was performed to obtain the joint posterior distribution of parameters using the Markov chain Monte Carlo (MCMC) method with the initial data generated in step (b) and the prior distributions. The MCMC sampling was performed using Stan via the rstan package version 2.21.7 [32] for R language version 4.1.3 [33]. For the MCMC sampling, 27,000 iterations were run for each of the four chains (with the first 2000 iterations excluded as a warm-up), resulting in a total of 100,000 iterations.

(d) WAIC was calculated for each sampling point of the design space using the joint posterior distribution of parameters, and the data sampling points were determined so that the sampling points having large WAIC would be preferably selected. The number of data sampling points was gradually increased to 1, 3, 5, 7, 10, 15, and 20. For comparison, data sampling points were also designed for conventional methods [15,17] using functional variance and empirical loss in addition to WAIC.

(e) New data were generated from the true model at the sampling points designed in step (d), and a Bayesian update of the predictive model was performed using the new data, the initial data, and the marginal prior distributions. For this Bayesian update, 2700 iterations of MCMC sampling were performed in each of the four chains (with the first 200 iterations excluded as a warm-up), and a total of 10,000 iterations were run. To account for the effect of randomness of the new data, the above Bayesian update was performed 100 times with different random number seeds when generating new data from the true model.

(f) The Bayes generalization loss, a measure of the model's prediction accuracy in the design space, was calculated using the joint posterior distribution of parameters obtained in step (e) to evaluate the predictive accuracy of the fracture probability estimation model.

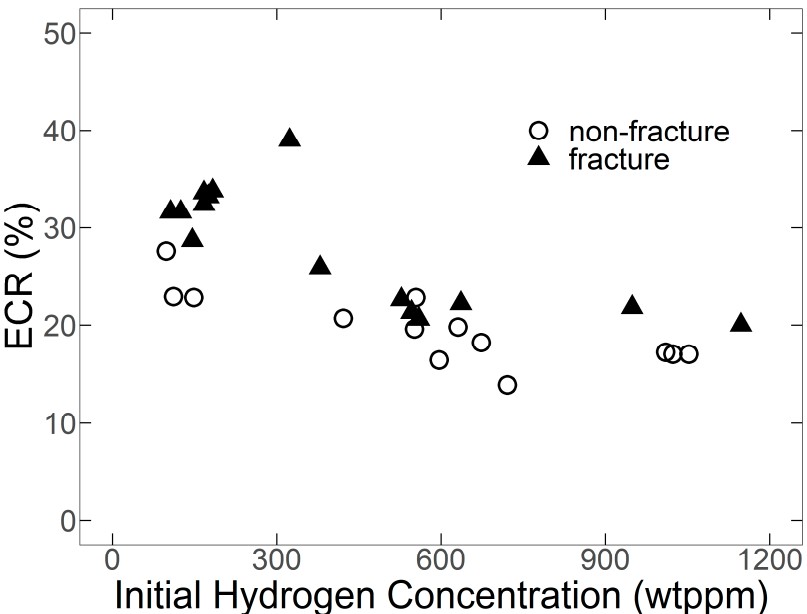

**Figure 2.** Initial data generated from the true model.

3.1.3. Results and Discussion

The relationship between the Bayes generalization loss and the number of data points added is shown in Figure 3. The results of 100 independent experiments with different random number seeds are shown in this figure as box plots, which follow a standard statistical convention: the boxes represent the interquartile range (IQR), with the lower and upper edges corresponding to the first and third quartiles, respectively. The whiskers extend to the most extreme values within 1.5 times the IQR from the quartiles, and values outside this range are considered outliers. For comparison, the results of conventional methods [15,17] using the functional variance and the empirical loss (information entropy) are also shown in this figure.

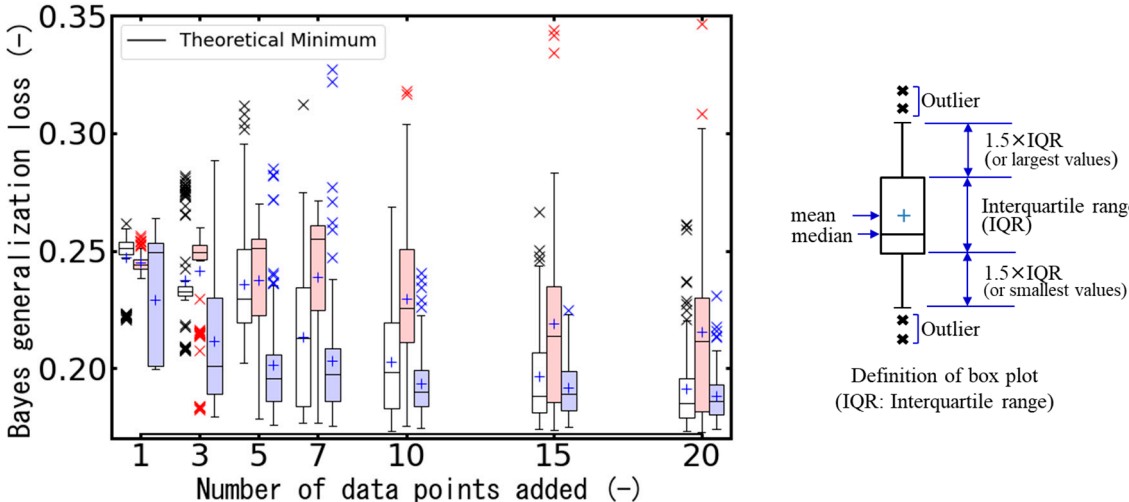

**Figure 3.** Relationship between the number of data added and the Bayes generalization loss. White: WAIC; red: functional variance; blue: empirical loss.

In this context, the Bayes generalization loss—defined as the expected KL divergence between the true data-generating process and the predictive distribution—serves as a proxy for epistemic uncertainty. A lower generalization loss indicates that the predictive distribution more closely approximates the true data-generating process, suggesting that

the model better reflects the underlying system and that epistemic uncertainty has been effectively reduced.

As shown in this figure, the Bayes generalization loss was lowest on average when the empirical loss was used, which demonstrates an effective reduction in epistemic uncertainty of fracture when using the empirical loss. Our proposed method showed a minimal reduction in generalization loss when the amount of additional data was small (fewer than ~10). However, as the number of additional data points increased, it tended to achieve a generalization loss comparable to that obtained using the empirical loss.

These results can be attributed to the consistent mathematical structure between the true model and the predictive model. Since the mathematical structures of the true model and the predictive model are identical, it is clear from their mathematical definitions that minimizing the empirical loss, which is the expected value of the negative log-likelihood of the predictive model, will also minimize the Bayesian generalization loss, which is the KL divergence between the true model and the predictive model.

Moreover, the relationship between the estimated parameters and the number of data points added is shown in Figure 4. As shown in this figure, the method using empirical loss is more capable of bringing the parameters closer to the true value than the proposed method when the amount of additional data is small (less than ~10 data points). However, as the number of additional data points increased, our proposed method approached the true value as well as the method using empirical loss. This shows that the proposed method accurately predicts the true values of the parameters.

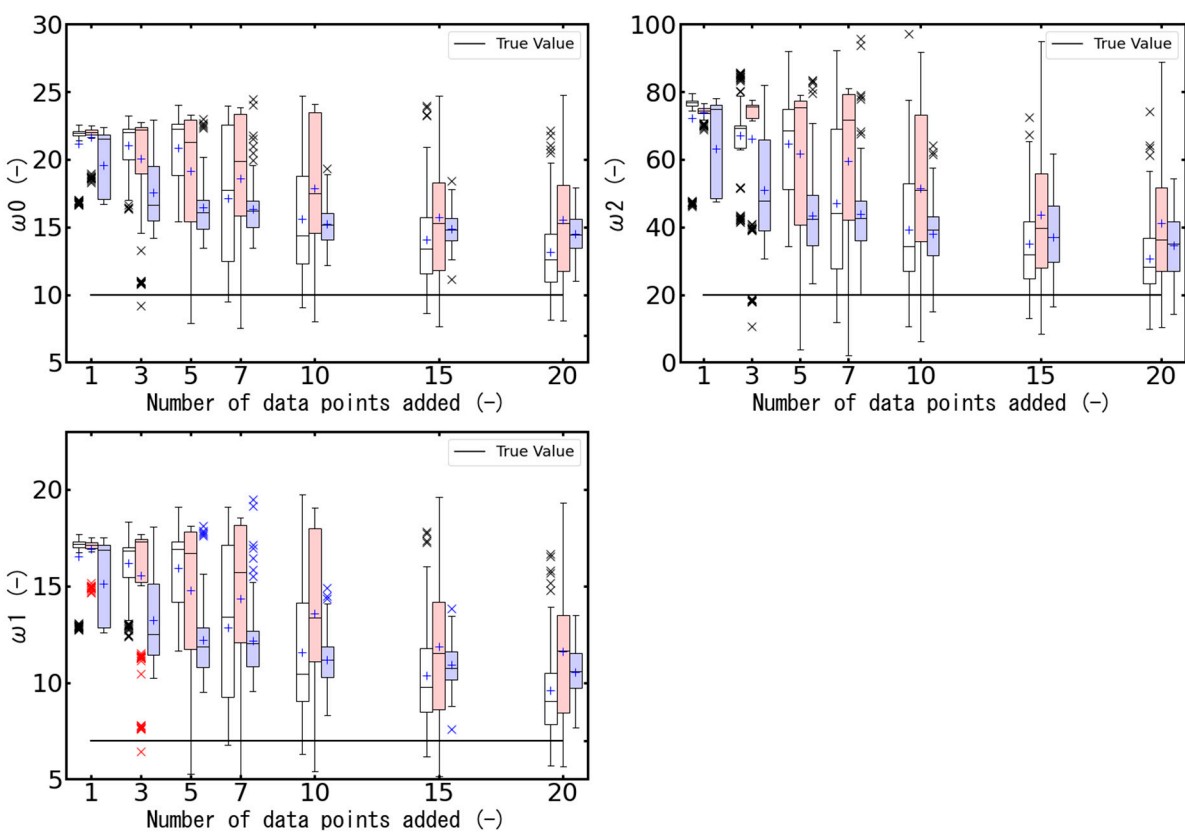

**Figure 4.** Relationship between the number of data added and each parameter values. White: WAIC; red: functional variance; blue: empirical loss.

## 3.2. Case 2: True and Predictive Models Have Different Mathematical Structures

### 3.2.1. Model Definition

The true model is defined as in Section 3.1.1.

In the real world, the true model cannot be known, and thus, the true model and the predictive model will not match. Therefore, in Case 2, a predictive model is defined as a model with a mathematical structure different from that of the true model. The following changes to the true model were applied to the predictive model:

- The standard normal cumulative distribution function $\Phi$ was changed to the logistic function.
- No logarithm was taken for explanatory variables.
- A cross-term was added for the explanatory variables.

Finally, the predictive model is defined as follows:

$$Y \sim Bernoulli(P(Y = 1|X)) \tag{14}$$

$$P(Y = 1|\mathbf{X}) = Logistic(\omega_0 + \omega_1 \tfrac{X_1}{100} + \omega_2 \tfrac{X_2}{10,000} + \omega_3 \tfrac{X_1 X_2}{10,000})$$
$$= \frac{1}{1 + \exp(-(\omega_0 + \omega_1 \tfrac{X_1}{100} + \omega_2 \tfrac{X_2}{10,000} + \omega_3 \tfrac{X_1 X_2}{10,000}))} \tag{15}$$

where $(\omega_0, \omega_1, \omega_2, \omega_3)$ are assumed to follow the following noninformative prior distribution:

$$\omega_k \sim Normal(0,\ 100)\ (k = 0, 1, 2, 3) \tag{16}$$

### 3.2.2. Calculation Flow

Numerical experiments were conducted with the same steps (a) to (f) in Section 3.1.2.

### 3.2.3. Results and Discussion

The relationship between the Bayes generalization loss and the number of data points added is shown in Figure 5. The results of 100 independent experiments with different random number seeds are shown in this figure as box plots, using the same definition as in Figure 3. For comparison, the results of conventional methods [15,17] using the functional variance and the empirical loss are also shown in this figure.

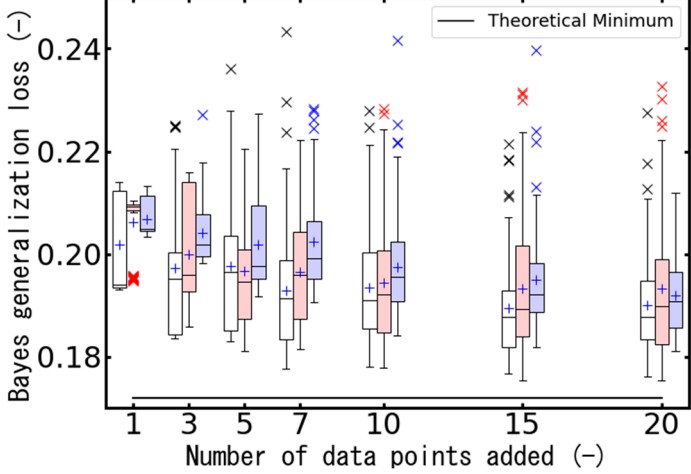

**Figure 5.** Relationship between the number of data added and the Bayes generalization loss. White: WAIC; red: functional variance; blue: empirical loss.

As shown in this figure, the Bayes generalization loss was lowest on average when WAIC was used to design experiments. Therefore, the proposed method is considered to effectively reduce the epistemic uncertainty of fracture when the number of experimental data is limited.

There were outliers regardless of the number of data points added for all the cases using WAIC, the functional variance, and the empirical loss. Upon investigating the

additional data in the cases where these outliers occurred, the binary data regarding fracture/non-fracture tended to differ from the binary predictions made by the predictive model. In other words, due to the probabilistic fluctuations of the samples, rare datasets that differed from the predictions were generated, resulting in a lack of improvement in prediction accuracy and a significant deterioration in the Bayes generalization loss.

## 4. Conclusions

This study aimed to develop a methodology to reduce epistemic uncertainty in fuel cladding fracture prediction under LOCA conditions, particularly in situations where experimental data are limited. Since the fracture limit of cladding tubes serves as a critical threshold in core damage determination in PRA, uncertainty in this parameter can significantly affect the realism of PRA. To address this, we proposed a Bayesian experimental design approach using an information criterion, WAIC, and conducted numerical experiments to evaluate its effectiveness.

Numerical experiments were conducted for the following two cases: one where the true model and the predictive model share the same mathematical structure (Case 1) and one where they have different mathematical structures (Case 2). In Case 1, when the number of newly added data points was relatively small (fewer than ~10), minimizing empirical loss, as proposed in a previous study, most effectively reduced the Bayes generalization loss, which is a measure of the accuracy of fracture predictions. This result aligns with the mathematical fact that minimizing empirical loss also minimizes the KL divergence between the true model and the predictive model. In Case 2, the proposed method using WAIC most effectively reduced the Bayesian generalization loss. Therefore, the proposed method enables more informative experimental designs on average and can reduce epistemic uncertainty in realistic situations where the true model is unknown. This indicates that the proposed method can predict fuel fracture stably with higher accuracy and less uncertainty, even when the experimental data are limited.

While the proposed method effectively reduces epistemic uncertainty, the Bayes generalization loss occasionally shows variability due to aleatory effects from random data generation. As future work, we will explore improvements such as using non-binary data, adopting sequential experimental design, and incorporating informative priors to enhance robustness and computational efficiency.

**Author Contributions:** Conceptualization, S.H., T.N. and T.T.; methodology, S.H., T.N. and T.T.; formal analysis, S.H.; data curation, S.H., T.N. and T.T.; writing—original draft preparation, S.H.; writing—review and editing, T.N. and T.T.; supervision, T.T.; project administration, T.N. and T.T.; funding acquisition, T.N. and T.T. All authors have read and agreed to the published version of the manuscript.

**Funding:** This work was supported by JSPS KAKENHI Grant Number JP23K13685.

**Data Availability Statement:** The datasets presented in this article are not readily available because the data include proprietary information. Requests to access the datasets should be directed to takata_t@n.t.u-tokyo.ac.jp.

**Conflicts of Interest:** The authors declare no conflicts of interest.

## Nomenclature

Abbreviations

| | |
|---|---|
| ECR | Equivalent cladding reacted |
| IQR | Interquartile range |
| LOCA | Loss-of-coolant accident |
| MCMC | Markov chain Monte Carlo |

| | |
|---|---|
| PRA | Probabilistic risk assessment |
| WAIC | Widely applicable information criterion |
| wtppm | Weight parts per million |
| Symbols | |
| $Bernoulli()$ | Bernoulli distribution |
| $E_\omega[]$ | The expectation value over the posterior distribution of $\omega$ |
| $E[]$ | The expected value |
| $G$ | Bayes generalization loss |
| $KL$ | Kullback–Leibler divergence between $q(y)$ and $p*(y)$ |
| $n$ | The number of samples. |
| $Normal()$ | Normal distribution |
| $p(y_i\|\omega)$ | Bayesian model |
| $P_{true}$ | Fracture probability estimated by the true model $q(y)$ |
| $P$ | Fracture probability estimated by the predictive model |
| $q(y)$ | True model |
| $S$ | Entropy |
| $T$ | Empirical loss |
| $V$ | Functional variance |
| $X_1$ | Explanatory variable for equivalent cladding reacted (ECR, %) |
| $X_2$ | Explanatory variable for the initial hydrogen concentration (wtppm) |
| $Y$ | LOCA-simulated test data binarized to 1 for fracture and 0 for non-fracture |
| $y_i$ | Random variable |
| $\omega$ | The vector of parameters |
| $(\omega_0, \omega_1, \omega_2)$ | Parameters to be estimated |
| $\Phi$ | Cumulative distribution function of the standard normal distribution |

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
