# Peer review of "A Bayesian Approach for Designing Experiments Based on Information Criteria to Reduce Epistemic Uncertainty of Fuel Fracture During Loss-of-Coolant Accidentsâ€"

_jne, doi:10.3390/jne6030035_

Round 1
Reviewer 1 Report
Comments and Suggestions for Authors
- remove self citation from the work. Only 1-2 papers can be accepted.
- The expanded version of the manuscript can be added in the supplementary section
- Key words should not contain abberviations
- Authors should have added nomenclatures and obberviations in the study
- Authors should add novelty but authors have added a simple line which is not good.
- https://doi.org/10.3390/inventions7030053 can refer to some relevant article
Author Response
Comment no. 1: remove self citation from the work. Only 1-2 papers can be accepted.
Response: We have limited self-citations to the minimum necessary—specifically, references [17], [19], [23], and [31] of the revised manuscript—as they are essential to the structure and context of this work. To address the concern, we have also incorporated a number of relevant external references ([7,8], [10-14], [20-22], and [24-33]) in the revised manuscript based on the editor’s and reviewers’ comments, thereby reducing the self-citation rate.
Comment no. 2: The expanded version of the manuscript can be added in the supplementary section
Response: We understand the suggestion; however, the revised manuscript is a self-contained and expanded version of our previous work. There is no additional material that needs to be included in the supplementary section.
Comment no. 3: Key words should not contain abberviations
Response: As suggested by the reviewer, we have replaced the keyword "LOCA" by "loss-of-coolant accident".
Comment no. 4: Authors should have added nomenclatures and obberviations in the study
Response: We have added a list of nomenclature and abbreviations to improve clarity on page 11, lines 360 to 394.
Comment no. 5: Authors should add novelty but authors have added a simple line which is not good.
Response: To clarify the novelty of our work, we have revised the Introduction in two ways. First, we have explicitly described the differences between the proposed WAIC-based design and existing entropy-based and functional variance-based approaches, emphasizing its robustness under model misspecification and limited data on page 3, lines 103 to 108 of the revised manuscript. Second, we have added a new paragraph summarizing the main contributions of the study to clearly present its novelty on page 3, lines 113 to 119 of the revised manuscript.
Comment no. 6: https://doi.org/10.3390/inventions7030053 can refer to some relevant article
Response: We sincerely appreciate the reviewer’s suggestion and have carefully reviewed the referenced work (https://doi.org/10.3390/inventions7030053). While it presents valuable insights, we determined that its focus and methodology differ from the specific scope of our study. For this reason, we respectfully chose not to include it in the revised manuscript.
Reviewer 2 Report
Comments and Suggestions for Authors
1. Quantitative results should be included in the abstract to help convey the significance and effectiveness of your approach.
2. The term epistemic uncertainty needs to be explained more clearly to engage a broader range of readers who may not be familiar with the concept.
3. Several related studies on the improvement of LOCA-simulated experiments must be cited. For example:
- Ref [1] modified heat transfer models in RELAP5 to improve predictions for LOCA accident experiments.
- Ref [2] proposed a new data assimilation framework for large-break LOCA test improvement.
- Ref [3] discussed enhancements of constitutive models related to reflooding to improve RELAP5 accuracy.
These are just a few examples; please consult the literature to find and include additional relevant references.
4. The sentence on lines 47–48 of page 2 (“..., methodologies that can extract more valuable information from a small number of experiments are needed.”) should be rewritten more clearly. Also, if there are established methods that address this challenge, please provide citations.
5. In lines 61–64, the sentence beginning with “While both entropy...” appears to highlight a limitation in previous studies. However, the logic is unclear. If something "does not necessarily" minimize KL divergence, is that alone a sufficient limitation? Please clarify the limitation in the existing literature that is more convincing.
6. The results in Figure 3 are difficult to interpret. For instance, if "+1.5×IQR = x" and the largest value is y, and "-1.5×IQR = a" and the smallest value is b, then what is displayed: in the downstream min(x, y) or max(x, y); min(a, b) or max(a, b) in the upstream? Please explain the plotting rule clearly, or consider an alternative way to present the uncertainties and improvements. Also, the absolute values of both the predictions and experimental results should be presented.
7. Please ensure that no new information is introduced in the Conclusion section. All points stated there should have been clearly presented and supported earlier in the paper.
References:
[1] Choi, Tong Soo, and Hee Cheon No. "Improvement of the reflood model of RELAP5/MOD3. 3 based on the assessments against FLECHT-SEASET tests." Nuclear Engineering and Design 240.4 (2010): 832-841.
[2] Tiep, Nguyen Huu, et al. "A newly proposed data assimilation framework to enhance predictions for reflood tests." Nuclear Engineering and Design 390 (2022): 111724.
[3] Li, Dong, Xiaojing Liu, and Yanhua Yang. "Improvement of reflood model in RELAP5 code based on sensitivity analysis." Nuclear Engineering and Design 303 (2016): 163-172.
Author Response
Comment no. 1: Quantitative results should be included in the abstract to help convey the significance and effectiveness of your approach.
Response: We have revised the abstract to include specific quantitative findings that demonstrate the effectiveness of the proposed method in reducing epistemic uncertainty on page 1, lines 24 to 32 of the revised manuscript.
Comment no. 2: The term epistemic uncertainty needs to be explained more clearly to engage a broader range of readers who may not be familiar with the concept.
Response: We have clarified the meaning of epistemic uncertainty in the Introduction by explaining its three components—parameter uncertainty, model uncertainty, and incompleteness uncertainty—and contrasting it with aleatory uncertainty on page 2, lines 37 to 60 of the revised manuscript.
Comment no. 3: Several related studies on the improvement of LOCA-simulated experiments must be cited. For example:  - Ref [1] modified heat transfer models in RELAP5 to improve predictions for LOCA accident experiments.
- Ref [2] proposed a new data assimilation framework for large-break LOCA test improvement.
- Ref [3] discussed enhancements of constitutive models related to reflooding to improve RELAP5 accuracy.
These are just a few examples; please consult the literature to find and include additional relevant references. 
References:
[1] Choi, Tong Soo, and Hee Cheon No. "Improvement of the reflood model of RELAP5/MOD3. 3 based on the assessments against FLECHT-SEASET tests." Nuclear Engineering and Design 240.4 (2010): 832-841.
[2] Tiep, Nguyen Huu, et al. "A newly proposed data assimilation framework to enhance predictions for reflood tests." Nuclear Engineering and Design 390 (2022): 111724.
[3] Li, Dong, Xiaojing Liu, and Yanhua Yang. "Improvement of reflood model in RELAP5 code based on sensitivity analysis." Nuclear Engineering and Design 303 (2016): 163-172.
Response: We thank the reviewer for the suggestion. The cited works have been reviewed and incorporated into the revised Introduction (page 3, lines 120 to 123) as relevant background and are now listed as refs. [20–22].
Comment no. 4: The sentence on lines 47–48 of page 2 (“..., methodologies that can extract more valuable information from a small number of experiments are needed.”) should be rewritten more clearly. Also, if there are established methods that address this challenge, please provide citations.
Response: We have revised the sentence to clarify the need for experimental design methodologies that can maximize the information obtained from limited experiments on page 2, lines 75 to 78 of the revised manuscript. We have also added citations to prior studies (Refs. [10–14]) that represent relevant established approaches.
Comment no. 5: In lines 61–64, the sentence beginning with “While both entropy...” appears to highlight a limitation in previous studies. However, the logic is unclear. If something "does not necessarily" minimize KL divergence, is that alone a sufficient limitation? Please clarify the limitation in the existing literature that is more convincing.
Response: We have revised the sentence to explain that the key issue lies in the lack of a principled mechanism in the conventional approaches to directly minimize the KL divergence between the true data-generating process and the predictive distribution on page 3, lines 92 to 99. Since KL divergence is a fundamental measure of epistemic uncertainty—quantifying the discrepancy between what the model predicts and what is true—failure to explicitly minimize it can lead to suboptimal experimental conditions, particularly under limited data or model misspecification.
Comment no. 6: The results in Figure 3 are difficult to interpret. For instance, if "+1.5×IQR = x" and the largest value is y, and "-1.5×IQR = a" and the smallest value is b, then what is displayed: in the downstream min(x, y) or max(x, y); min(a, b) or max(a, b) in the upstream? Please explain the plotting rule clearly, or consider an alternative way to present the uncertainties and improvements. Also, the absolute values of both the predictions and experimental results should be presented.
Response: To clarify the plotting rule, we have added explanations on page 7, line 253 to page 8, line 257, and page 10, lines 316–317. The box plots follow the standard definition: the box spans from the first quartile (Q1) to the third quartile (Q3), with the median and mean marked inside. The whiskers extend to the most extreme data points within 1.5×IQR (interquartile range, Q3–Q1) from Q1 and Q3. Specifically, the upper whisker reaches up to min(y, Q3 + 1.5×IQR), and the lower whisker down to max(b, Q1 – 1.5×IQR), where y and b are the maximum and minimum observed values, respectively. Values outside this range are shown as outliers.
As for the absolute values of predictions and experimental results, we respectfully note that the focus of Figure 3 is on the relative comparison of epistemic uncertainty reduction among different methods. Since our analysis aims to evaluate comparative performance rather than report absolute predictive values, we believe the current representation is sufficient for interpretation.
Comment no. 7: Please ensure that no new information is introduced in the Conclusion section. All points stated there should have been clearly presented and supported earlier in the paper.
Response: As suggested by the reviewer, we have revised the Conclusion section to avoid introducing new information on page 11, lines 354 to 358. The discussion of future work has been simplified to ensure that it only summarizes and builds upon points already presented and supported in the main text.
Reviewer 3 Report
Comments and Suggestions for Authors
1) to this reviewer, the investigation appears limited to a data treatment methodology or approach, that in spite of its complexity, works in the two cases -Case 1 and 2 given.
2) unfortunately, only non-irradiated fracture is addressed when most interest is in irradiated cladding.
3) Q1. What is the dominant and/or ranked influential factors in fracture under irradiated conditions?
4) Q2. Was the method validate and verified (or confirmed) in some manner in say brittle fracture under controlled conditions? Q3. For instance, what is the hypothesized condition of cladding of a given porosity if subject to thousands of cycled dryout (film boiling) conditions.
5) Q4. What is the known or postulated impact of a Markovian versus a non-Markovian approach? Q5. Was this investigated?
6) in this reviewer's opinion, some aspect of "certainties" and "uncertainties" in LOCA-initiated fracture needs to be discussed.
7) past works of M. Ishii, I. Kataoka and co-authors may reveal the time-scales and additional details of LOCA-induced cladding fracture failure. Q6. what is the postulated cladding failure mechanism under which (limited) experiments have been conducted?
Author Response
Comment no. 1: to this reviewer, the investigation appears limited to a data treatment methodology or approach, that in spite of its complexity, works in the two cases -Case 1 and 2 given.
Response: We have revised the introductory paragraph of Section 3 to clarify that the two cases (Case 1 and Case 2) were selected as representative scenarios to evaluate the proposed method under both ideal and practical conditions, including model misspecification on page 5, line 187 to page 6, line 199. While the numerical experiments focus on these two cases, the proposed Bayesian design framework is general and applicable to a broader class of problems in safety-critical domains. The revisions help emphasize that the methodology is not limited to these cases.
Comment no. 2: unfortunately, only non-irradiated fracture is addressed when most interest is in irradiated cladding.
Response: We have revised the manuscript to clarify that while the current numerical experiments use a model for non-irradiated cladding, our previous study (Ref. [31]) has demonstrated that the proposed method is also applicable to high-burnup cladding tubes on page 5, lines 181 to 186 of the revised manuscript. This highlights the broader relevance of our approach to realistic safety evaluation contexts.
Ref. [31]: Narukawa, T.; Amaya, M. Fracture limit of high-burnup advanced fuel cladding tubes under loss-of-coolant accident conditions. J. Nucl. Sci. Technol. 2019, 57, 68–78.
Comment no. 3: Q1. What is the dominant and/or ranked influential factors in fracture under irradiated conditions?
Response: Our previous study (Ref. [31]) has shown that Equivalent Cladding Reacted (ECR) and initial hydrogen concentration are the dominant factors influencing fracture under irradiated conditions. Since these variables are explicitly included in the fracture probability model used in this study, it is applicable not only to unirradiated but also to irradiated cladding, as described in our response to Comment 2 of the reviewer 3.
Ref. [31]: Narukawa, T.; Amaya, M. Fracture limit of high-burnup advanced fuel cladding tubes under loss-of-coolant accident conditions. J. Nucl. Sci. Technol. 2019, 57, 68–78.
Comment no. 4: Q2. Was the method validate and verified (or confirmed) in some manner in say brittle fracture under controlled conditions? Q3. For instance, what is the hypothesized condition of cladding of a given porosity if subject to thousands of cycled dryout (film boiling) conditions.
Response: The fracture probability model used in this study is based on experimental data obtained under conditions that simulate actual LOCA scenarios. It does not consider phenomena such as high-cycle dryout or cladding porosity evolution, as these are outside the scope of the current model and its intended application.
Comment no. 5: Q4. What is the known or postulated impact of a Markovian versus a non-Markovian approach? Q5. Was this investigated?
Response: In this study, Markov Chain Monte Carlo (MCMC) methods were used for Bayesian parameter estimation and WAIC calculation. All chains exhibited good convergence, and the posterior estimates were stable. Therefore, differences between Markovian and non-Markovian approaches do not affect the conclusions of this work.
Comment no. 6: in this reviewer's opinion, some aspect of "certainties" and "uncertainties" in LOCA-initiated fracture needs to be discussed.
Response: To clarify the nature of uncertainty in LOCA-initiated fracture, we have supplemented the Introduction with a description of the underlying factors contributing to the uncertainty in the fracture limit of fuel cladding tubes on page 2, lines 62 to 71 of the revised manuscript. In particular, we now explain that this uncertainty stems from both the difficulty in precisely estimating the thermal and mechanical stresses induced during quenching and the degradation of cladding strength due to oxidation, hydrogen embrittlement, and ballooning. This clarification helps to articulate the type of epistemic uncertainty addressed in this study.
Comment no. 7: past works of M. Ishii, I. Kataoka and co-authors may reveal the time-scales and additional details of LOCA-induced cladding fracture failure. Q6. what is the postulated cladding failure mechanism under which (limited) experiments have been conducted?
Response: While the contributions of M. Ishii, I. Kataoka, and others are indeed foundational in the field of two-phase flow and system thermal-hydraulics, our study focuses on modeling fuel cladding fracture under LOCA conditions, where the dominant failure mechanisms are cladding embrittlement due to oxidation and hydrogen uptake. These mechanisms have been extensively established through decades of experimental research (Refs. [24–31]), and the dataset used in our study is based on LOCA simulation tests that directly incorporate these phenomena. To clarify this point, we have added an explanatory sentence in Section 3 (on page 5, lines 177 to 186) of the revised manuscript.
References:
- Chung, H.M. Fuel behavior under loss-of-coolant accident situations. Nucl. Eng. Technol. 2005, 37, 327–362.
- Meyer, R.O. Fuel Behavior under Abnormal Conditions; U.S. Nuclear Regulatory Commission: Washington, DC, USA, 2013; Report No. NUREG/KM-0004.
- Uetsuka, H.; Furuta, T.; Kawasaki, S. Failure-bearing capability of oxidized Zircaloy-4 cladding under simulated loss-of-coolant condition. J. Nucl. Sci. Technol. 1983, 20, 941–950.
- Nagase, F.; Fuketa, T. Effect of pre-hydriding on thermal shock resistance of Zircaloy-4 cladding under simulated loss-of-coolant accident conditions. J. Nucl. Sci. Technol. 2004, 41, 723–730.
- Nagase, F.; Fuketa, T. Behavior of pre-hydrided Zircaloy-4 cladding under simulated LOCA conditions. J. Nucl. Sci. Technol. 2005, 42, 209–218.
- Nagase, F.; Fuketa, T. Fracture behavior of irradiated Zircaloy-4 cladding under simulated LOCA conditions. J. Nucl. Sci. Technol. 2006, 43, 1114–1119.
- Cabrera, A.; Waeckel, N. A strength based approach to define LOCA limits. In Proceedings of TopFuel 2015; Zurich, Switzerland, 14–16 September 2015.
- Narukawa, T.; Amaya, M. Fracture limit of high-burnup advanced fuel cladding tubes under loss-of-coolant accident conditions. J. Nucl. Sci. Technol. 2019, 57, 68–78.
Round 2
Reviewer 1 Report
Comments and Suggestions for Authors
Accept
Reviewer 2 Report
Comments and Suggestions for Authors
Thank you for your response.